# Relational Concept Bottleneck Models

**Pietro Barbiero**
Università della Svizzera Italiana
University of Cambridge
barbiero@tutanota.com

**Francesco Giannini**
Scuola Normale Superiore
francesco.giannini@sns.it

**Gabriele Ciravegna**
Politecnico di Torino
gabriele.ciravegna@polito.it

**Michelangelo Diligenti**
University of Siena
michelangelo.diligenti@unisi.it

**Giuseppe Marra**
KU Leuven
giuseppe.marra@kuleuven.be

## Abstract

The design of interpretable deep learning models working in relational domains poses an open challenge: interpretable deep learning methods, such as Concept Bottleneck Models (CBMs), are not designed to solve relational problems, while relational deep learning models, such as Graph Neural Networks (GNNs), are not as interpretable as CBMs. To overcome these limitations, we propose Relational Concept Bottleneck Models (R-CBMs), a family of relational deep learning methods providing interpretable task predictions. As special cases, we show that R-CBMs are capable of both representing standard CBMs and message-passing GNNs. To evaluate the effectiveness and versatility of these models, we designed a class of experimental problems, ranging from image classification to link prediction in knowledge graphs. In particular we show that R-CBMs (i) match generalization performance of existing relational black-boxes, (ii) support the generation of quantified concept-based explanations, (iii) effectively respond to test-time interventions, and (iv) withstand demanding settings including out-of-distribution scenarios, limited training data regimes, and scarce concept supervisions.

## 1  Introduction

Chemistry, politics, economics, traffic jams: we constantly rely on relations to describe, explain and reason on everyday life problems. For instance, we can easily deduce Bart's citizenship if we consider Homer's citizenship and his status as Bart's father (Figure 1). While relational Deep Learning (DL) models [29, 19, 32, 16] can effectively solve such problems, the design of *interpretable* neural models capable of relational reasoning is still an open challenge. Among DL methods, Concept Bottleneck Models (CBMs) [13] are interpretable methods explaining their predictions by first mapping input features to a set of human-understandable concepts and

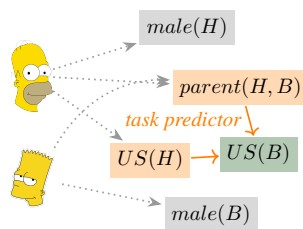

Figure 1: Relational Concept Bottleneck Models can correctly predict and explain Bart's (B) citizenship by considering Homer's (H) citizenship and his status as Bart's parent.

38th Conference on Neural Information Processing Systems (NeurIPS 2024).

then using such concepts to solve the given tasks. However, current CBMs are not well-suited for addressing relational problems as they can process only one input entity at a time by construction. To solve relational problems, CBMs would need to handle concepts/tasks involving multiple entities (e.g., the concept "parent" which depends on both the entity "Homer" and "Bart"), thus forcing CBMs to process more entities at a time. Moreover, the definition of a suitable relational bottleneck layer is generally not straightforward, as a task prediction may require complex connections among multiple relational concepts. On the other side, while existing relational DL methods, such as Graph Neural Networks (GNN), may effectively solve such problems (e.g., correctly predicting Bart's citizenship), they are still unable to explain their predictions as CBMs would do (e.g., Bart is a US citizen *since Homer is a US citizen and Homer is the father of Bart*). Hence, a knowledge gap persists in the existing literature: defining a DL model capable of relational reasoning (akin to a GNN), while also being interpretable (akin to a CBM).

To address this gap, we propose Relational Concept Bottleneck Models (R-CBMs, Section 3), a family of concept bottleneck models where both concepts and tasks may depend on multiple entities, and that have both CBMs and GNNs as special cases. The results of our experiments (Section 4 and 5) show that R-CBMs: (i) match the generalization performance of existing relational black-boxes, (ii) support the generation of first-order logic explanations, (iii) effectively respond to test-time concept and rule interventions improving their task performance, (iv) withstand demanding test scenarios including out-of-distribution settings, limited training data regimes, and scarce concept supervisions.

## 2  Background

**Concept bottleneck models.** A Concept Bottleneck Model (CBM) is a function composing: (i) a concept encoder $g : X \to C$ mapping each entity $e$ with feature representation $x_e \in X \subseteq \mathbb{R}^d$ (e.g., an image) to a set of $k$ concepts $c \in C \subseteq [0, 1]^k$ (e.g., "red","round"), and (ii) a task predictor $f : C \to Y$ mapping concepts to a set of $m$ tasks $y \in Y \subseteq [0, 1]^m$ (e.g., "apple","tomato"). Each component $g_i$ and $f_j$ vehicle the prediction of the $i$-th concept and $j$-th task, respectively.

**Relational languages.** A relational setting can be outlined using a function-free first-order logic language $\mathcal{L} = (\mathcal{E}, \mathcal{V}, \mathcal{P})$, where $\mathcal{E}$ is a finite set of constants for specific domain entities[1], $\mathcal{V}$ is a set of variables for anonymous entities, and $\mathcal{P}$ is a set of $n$-ary predicates for relations among entities. The central objects of a relational language are its *atoms*, i.e. expressions $p(\tau_1, \ldots, \tau_n)$, where $p$ is an $n$-ary predicate and $\tau_1, \ldots, \tau_n$ are constants or variables. In case $\tau_1, \ldots, \tau_n$ are all constants, $p(\tau_1, \ldots, \tau_n)$ is called a *ground atom*. Examples of atoms can be *male*(*Bart*) and *parent*(*u, v*), with $Bart \in \mathcal{E}$ and $u, v \in \mathcal{V}$. Given a set of atoms $\Gamma$ defined on a joint set of variables $V = \{v_1, \ldots, v_n\}$, the process of applying a substitution $\theta_V = \{v_1/e_1, ..., v_n/e_n\}$ to $\Gamma$ is called *grounding*, i.e. the substitution of all the variables $v_i$ with some constants $x_i$, according to $\theta_V$. For example, given $\Gamma = [parent(v_1, v_2), parent(v_2, v_3)]$ and the substitution $\theta = \{v_1/Abe, v_2/Homer, v_3/Bart\}$, we can obtain the ground list $\theta\Gamma = [parent(Abe, Homer), parent(Homer, Bart)]$. The set of all the ground atoms of a relational language is called its Herbrand base (*HB*). Logic rules are defined as usual by applying logic connectives $\{\neg, \wedge, \vee, \rightarrow\}$ and quantifiers $\{\forall, \exists\}$ on atoms.

**Graph neural networks.** The architecture of a typical GNN for node-classification tasks consists of three primary steps. For every node $i$, 1) an incoming message $M_{j \to i}$ is passed from a neighbor node $j \in \mathcal{N}(i)$ to $i$, where $\mathcal{N}(i)$ denotes the set of all the incoming neighbours of $i$, 2) the embedding representation of node $i$ is updated by aggregating all the incoming messages from its neighbors, 3) a readout function is applied to the node embeddings to predict the class label $\hat{y}(i)$. Steps 1)-2) are typically repeated multiple times to allow multi-hop information propagation.

## 3  Relational Concept Bottleneck Models

This work addresses a key research question: *how can we bridge the gap between the interpretability of concept-based models and the reasoning capabilities of relational DL?* To answer this question, we extend the notion of bottleneck to a relational setting (Section 3.1) and classic message-passing to also update atom predictions during the recursive steps (Section 3.2). Then we illustrate the learning problem that R-CBMs can solve (Section 3.3), and finally we discuss the connections of R-CBMs with both standard CBMs and GNNs (Section 3.4).

---

[1]Assuming a 1-to-1 mapping between constants and entities allows us to use these words interchangeably.

### 3.1 Relational Concept Bottlenecks

The structure of the relational concept bottleneck can be defined as a relational structure, where each atom corresponds to a node. The dependencies among the atoms are represented as a directed hypergraph, where each hyperedge is positional, and can have multiple nodes as head, but only a single node as tail. In this regard, every hyperedge defines a relational concept bottleneck from (possibly) many source ground atoms to a destination ground atom.

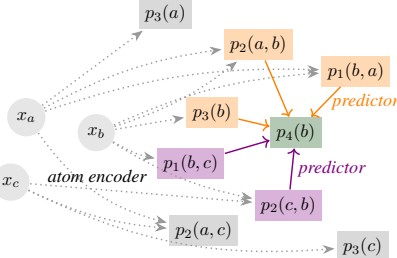

Moreover, each hyperedge is assumed to be sufficient to carry out the prediction for the destination atom, which can however be collectively improved by merging separate predictions. Formally, an *atom dependency graph* for a node $A$, is a positional, labeled hypergraph $\mathcal{H} = (HB, \mathcal{R})$, whose nodes are the atoms in the Herbrand Base *HB* of a relational language, and each hyperedge $r \in \mathcal{R}$ is such that $r = ([A_1, \ldots, A_m], [A])$, with $A_1, \ldots, A_m, A \in HB$, meaning that in $\mathcal{H}$ there is a hyperedge with source $[A_1, \ldots, A_m]$ and destination $A$. Each hyperedge is labelled with a type identifier $l(r)$. Given an atom $A$, we indicate by $\mathcal{R}(A)$ the set of hyperedges with destination $A$, and by $\mathcal{N}_r(A)$ the source of the hyperedge $r$ if $r \in \mathcal{R}(A)$ or the empty­set otherwise. Figure 2 shows an example with two hy­peredges with destination $p_4(b)$, where $\mathcal{N}_{orange}(p_4(b)) = [p_3(b), p_2(a,b), p_1(b,a)]$ and $\mathcal{N}_{violet}(p_4(b)) = [p_1(b,c), p_2(c,b)]$, we used different colors to iden­tify different hyperedges.

Figure 2: The graph represents the de­pendencies among the atoms. Here, the atom $p_4(b)$ can be predicted either from the orange $[p_3(b), p_2(a,b), p_1(b,a)]$ or violet $[p_1(b,c), p_2(c,b)]$ tuples of neighbours.

### 3.2 The Model

Relational Concept Bottleneck Models (R-CBM) merge CBMs and GNNs into an interpretable relational setting. An R-CBM first processes the atoms of a relational language by an encoder, and then map them by a predictor (like a CBM). The final prediction is computed by aggregating all the ones from separate groups of neighbour atoms, according to a given dependency graph (like a GNN). The pipeline of R-CBMs can be described as follows: (i) the atom encoder and predictor embeds each atom into a concept embedding and prediction score, respectively (ii) message-passing is performed to refine the embeddings and scores according to the structure defined by the *atom dependency graph*, and (iii) the atom predictions are obtained by *aggregating* the predictions.

**Atom encoder.** An $n$-ary ground atom $A$ is defined by a predicate $p$ and a tuple of entities $\mathbf{e} = (e_1, \ldots, e_n)$, such that $A = p(\mathbf{e})$. Like in a GNN, the entities have a feature representation $\mathbf{x_e} = (x_{e_1}, \ldots, x_{e_n}) \in \mathbb{R}^{d \cdot n}$, being $d$ the representation size. For each atom $A = p(\mathbf{e})$, the atom encoder $g_p$ computes the atom encoding $g_p(\mathbf{x_e}) \in \mathbb{R}^H$, being $H$ the embedding size.

**Message-passing.** Given the relational concept bottlenecks, the updating of the embeddings and predictions of the atoms can be expressed as a message-passing schema over the dependency graph. For each $A \in HB$, with $A = p(\mathbf{e})$, the initial embedding and prediction for $A$ are calculated by:

$$h^0(A) = g_p(\mathbf{x_e}), \qquad y^0(A) = s(h^0(A))$$

where $g_p$ is the atom encoder and $s : \mathbb{R}^H \to [0, 1]$ is a learnable predictor function working on the local (non-relational) embeddings, such as an MLP with sigmoid activation function. Assuming the message-passing is running for $T$ time steps, for every $r \in \mathcal{R}(A)$, $1 \le t \le T$, we have the updates:

$$h_r^t(A) = u_{l(r)}\left(h^{t-1}(A), \left[h^{t-1}(B)\right]_{B \in \mathcal{N}_r(A)}\right)$$
$$y_r^t(A) = f_{l(r)}\left(y^{t-1}(A), \left[h_r^t(B), y^{t-1}(B)\right]_{B \in \mathcal{N}_r(A)}\right)$$
$$h^t(A) = \sum_{r \in \mathcal{R}(A)} h_r^t(A)$$
$$y^t(A) = \bigoplus_{r \in \mathcal{R}(A)} y_r^t(A)$$

where $u_{l(r)}$ and $f_{l(r)}$ are edge-type specific functions implementing, respectively: a combine/update operation that provides a refined latent representation $h_r^t(A)$, and a local readout operation that provides a candidate prediction based on a single neighbourhood $h_r^t(A)$. The operator $\bigoplus$ aggregates the predictions over all the neighbourhoods $r \in \mathcal{R}(A)$, e.g, by maximum or summation, whose selection criterion for interpretable models will be discussed in Section 3.4.

**Relational task predictor.** The final task prediction for the atom $A$ is given by $y^T(A)$, via the combination of the aggregation function and the local readout $f_{l(r)}$ at time $T$. We note that this formulation unifies and extends the role of the task predictor in CBMs and of a readout function in GNNs. In practice, each task predictor $f_{l(r)}$ can be implemented by any blackbox-like function, like the one used in GNN architectures [29] or **CBM-Deep** [13]. However, an interesting alternative is to use either a partially interpretable function, like in **linear CBMs**, or a fully interpretable function, like in Deep Concept Reasoners (**DCR**) [1], which constructs a logic rule combining the predictions of the incoming atoms. Further details on selectable task predictors are in App. A.4.

**Example 3.1.** Given a local neighbourhood for the atom *grandparent*(*Abe*, *Bart*), such that

$$\mathcal{N}_r(grandparent(Abe, Bart)) = [parent(Abe, Homer), parent(Homer, Bart), parent(Homer, Lisa)]$$

a non-interpretable $f_{l(r)}$ can compute the prediction $y_r^T(grandparent(Abe, Bart))$ based on the neighbourhood. An interpretable $f_{l(r)}$, like the one used by DCR, can also provide an explanation for the prediction, like e.g. *parent*(*Abe*, *Homer*) $\land$ *parent*(*Homer*, *Bart*) $\rightarrow$ *grandparent*(*Abe*, *Bart*).

### 3.3 Learning

In this paper, we use a joint (end-to-end) SGD training of the atom encoder and predictor, as the original CBM paper [13] suggests for generalization. The learning problem can be stated as follows.

**Definition 3.2** (Learning Problem). *Given:* a relational language $(\mathcal{E}, \mathcal{V}, \mathcal{P})$ with all the atoms collected in *HB*; a set of entities represented by their corresponding feature vectors in $X$ (i.e. *the input*); a dataset composed of a subset of supervised atoms $D = \{(A_i, l_i) : A_i \in HB, l_i \in \{0, 1\}\}$, where $l_i$ is the corresponding ground-truth value for $A_i$; models $g_p, s, u_{l(r)}, f_{l(r)}$ with parameters $\pi$ and a maximum number of iterations $T$; an atom dependency graph determining the relational structure of all the atoms; a loss function $L$. *Find:* $\min_\pi \sum_{(A_i, l_i) \in D} L(y^T(A_i), l_i)$.

### 3.4 Examples of Relational Concept Bottlenecks

To derive specific instantiations of R-CBMs, we introduce the notion of *templetized hyperedge* for an atom dependency graph. A templetized hyperedge $\rho$ is defined as a standard hyperedge, but where the source and/or the destination contain one or more variables. For example, $\rho = ([p_1(v, u), p_2(u, e)], p(v, e))$ is a templetized hyperedge, meaning that we have an hyperedge istance for each possible grounding to the variables occurring in the atoms (i.e. $v, u$ in the example). All the hyperedges generated by the same template are associated to the same edge label $l(r)$, so that both model functions $u_{r(l)}, f_{r(l)}$, are shared across all instances of the same template. We omit the subscript in case there is a single templetized hyperedge in the hypergraph (e.g. one single edge type in the dependency graph).

**Case #1: Standard CBMs.** A standard (non-relational) CBM can be easily seen as an R-CBM, by making few assumptions on the relational language and the atom dependency graph it is based on: (i) all predicates are unary, and can be partitioned into two disjoint sets, i.e. the concepts $c_1, \ldots, c_k$ and tasks $t_1, \ldots, t_m$, respectively, (ii) in the atom dependency graph any concept atom has no parents, i.e. for every atom of the form $c(\mathbf{e})$, we have $\mathcal{R}(c(\mathbf{e})) = \emptyset$, and for every task $t$ there is exactly one templetized hyperedge $\rho$ whose source is composed by all the concept atoms, i.e. $\mathcal{N}_\rho(t(v)) = [c_1(v), \ldots, c_k(v)]$, (iii) $T = 1$. Hence, the final prediction on an atom $A = p(\mathbf{e})$ is obtained as $y(A) = s(g_p(\mathbf{x_e}))$ if $p$ is a concept predicate, and $y(A) = f_\rho(y(c_1(\mathbf{e})), \ldots, y(c_k(\mathbf{e})))$, if $p$ is a task predicate. For instance, given the unary predicates $\mathcal{P} = \{red, round, tomato\}$, a possible concept bottleneck is given by the templetized hyperedge $\mathcal{N}(tomato(v)) = [red(v), round(v)]$.

**Case #2: Node classification via GNNs.** R-CBMs allow the modelling of simple relational structures, such as relation-entity graphs, which are typically used by GNNs to generate and update node embeddings. For instance, let us consider a node classification task wrt a class $p$, and let $q$ denotes the relation in the graph. This can be represented by the templetized hyperedge $\mathcal{N}(p(v)) = [q(v, u)]$. For a heterogeneous graph with $q_1, \ldots, q_k$ relations, we can instead consider the templetized hyperedge $\mathcal{N}(p(v)) = [q_1(v, u), \ldots, q_k(v, u)]$. Message-passing and readout in GNNs are special cases of R-CBMs, as embeddings do not depend on atom predictions and readout occurs only at step $T$:

$$\begin{aligned} h^t(A) &= u\left(h^{t-1}(A), \left[h^{t-1}(B)\right]_{B \in \mathcal{N}(A)}\right) \\ y^T(A) &= f\left(h^T(A)\right) \end{aligned}$$

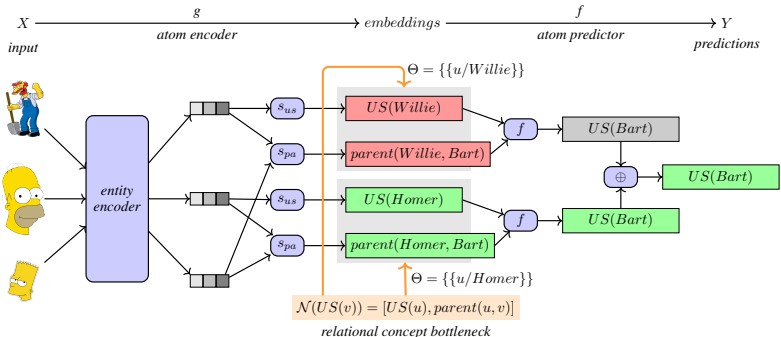

Figure 3: In R-CBMs (i) the atom encoder $g$ maps input entities to a set of ground atoms (red/green indicate the ground atom label false/true), (ii) the relational bottleneck guides the selection of concept atoms by considering all the possible variable substitutions in $\Theta$, (iii) the atom predictor $f$ maps the selected atoms into a task prediction, and (iv) the aggregator $\oplus$ combines all evidence into a final task prediction.

**Case study #3: Templetized relational concept bottlenecks.** The templatization of standard CBMs can be further generalized, as relational bottlenecks can represent more complex interactions.

**Definition 3.3** (Templetized relational concept bottleneck)**.** Given an $n$-ary predicate $p$, and an integer $w \geq 0$, we define a *templetized relational concept bottleneck* of width $w$ as the expression:

$$\mathcal{N}(p(\bar{v})) = b(\bar{v}, \bar{u}) \tag{1}$$

where $\bar{v} = (v_1, \ldots, v_n)$, $\bar{u} = (u_1, \ldots, u_w)$ are variables and $b(\bar{v}, \bar{u})$ is a list of atoms with predicates in $\mathcal{P}$ and tuples of variables taken from $\{v_1, \ldots, v_n, u_1, \ldots, u_w\}$.

For instance, assuming to partition the predicates into two disjoint sets, i.e. concepts and tasks, similarly to what considered for standard CBMs, Definition 3.3 specifies the input-output interface of a concept-based task predictor in a relational context, being $p$ a task predicate and the predicates contained in $b$ the concepts[2]. The following example grounds this definition in a concrete setting.

**Example 3.4.** Given the binary predicates *grandparent* (task) and *parent* (concept), $w = 1$ and $b(v_1, v_2, u) = [parent(v_1, u), parent(u, v_2)]$ we get the templetized relational concept bottleneck:

$$\mathcal{N}(grandparent(v_1, v_2)) = [parent(v_1, u), parent(u, v_2)]$$

Figure 3 illustrates the instantiation of a relational concept bottleneck $\mathcal{N}(US(v)) = [US(u), parent(u, v)]$, for $\mathcal{E} = [Willie, Homer, Bart]$.

We notice that by replacing the variables $\bar{v}$ with an entity tuple in Definition 3.3 does not correspond to an univocal instantiation of a relational concept bottleneck, as the same destination atom is predicted for every substitution $\theta$ of the variables $\bar{u}$. For instance if $\mathcal{E} = [Abe, Homer, Bart, Lisa]$ in Example 3.4, we have different hyperedges having $grandparent(Abe, Bart)$ as tail:

$$\mathcal{N}(grandparent(Abe, Bart)) = [parent(Abe, Abe), parent(Abe, Bart)] \qquad (\theta_u = \{u/Abe\})$$
$$\mathcal{N}(grandparent(Abe, Bart)) = [parent(Abe, Homer), parent(Homer, Bart)] \qquad (\theta_u = \{u/Homer\})$$
$$\mathcal{N}(grandparent(Abe, Bart)) = [parent(Abe, Bart), parent(Bart, Bart)] \qquad (\theta_u = \{u/Bart\})$$
$$\mathcal{N}(grandparent(Abe, Bart)) = [parent(Abe, Lisa), parent(Lisa, Bart)] \qquad (\theta_u = \{u/Lisa\})$$

Each separate grounding of a relational concept bottleneck corresponds to a separate predicate prediction, as the same destination atom can be predicted by different bottlenecks. Taking again Example 3.4, we can assume to have also the bottleneck: $\mathcal{N}_2(grandparent(v_1, v_2)) = [grandparent(v_1, u), sister(u, v_2)]$ which also adds the hyperedge:

$$\mathcal{N}_2(grandparent(Abe, Bart)) = [grandparent(Abe, Lisa), sister(Lisa, Bart)] \quad (\theta_u = \{u/Lisa\})$$

The final prediction is obtained by aggregating all the single predictions with the $\bigoplus$ operator.

---

[2]App. 3 discusses how to select the atoms input to a relational concept bottleneck.

**Aggregation semantics.** In standard CBMs the interpretation of the prediction solely depends on the task predictor $f$ and there is a single templetized hyperedge from concepts to any task node. However, the same atom can be predicted by different bottlenecks in R-CBMs. Indeed, different bottlenecks can represent separate dependency paths for the same atom like: $\mathcal{N}_1(t(\bar{v})) = b_1(\bar{v}, \bar{u}_1), \mathcal{N}_2(t(\bar{v})) = b_2(\bar{v}, \bar{u}_2), \ldots$ Even when considering a single bottleneck, the same grounding for $\bar{v}$ corresponds to multiple dependencies for the same atom if $w > 0$. Each ground bottleneck corresponds to a separate hyperedge in the dependency graph and it plays a fundamental role the choice of the aggregation function $\bigoplus$. In this paper, we select as $\bigoplus = \max$, as it guarantees a sound interpretation to R-CBMs' predictions. Indeed, the $\max$ aggregation corresponds to the semantics of an *existential quantification* on the variables $\bar{u}$. As a result, the final task prediction is true if the task predictor $f$ fires for at least one grounding of the extra variables.

**Example 3.5.** Following Example 3.4, we consider a task predictor $f$ as a logic conjunction ($\wedge$) between concept atoms. If we use $\bigoplus = \max$, then the final task prediction is true if at least one substitution for $u$ is true, i.e. if there exists an entity that is parent of *Bart* and such that *Abe* is her/his parent. Hence, the final task prediction can be interpreted as the logic formula

$$\exists u \, parent(Abe, u) \wedge parent(u, Bart) \rightarrow grandparent(Abe, Bart)$$

In summary, assuming $\bigoplus = \max$ and that each $f$ is realized as a logic rule $\varphi$, a relational concept bottleneck with $\mathcal{N}(p(\bar{v})) = b(\bar{v}, \bar{u})$ can be associated with the explanation:

$$\forall \bar{v} \, \exists \bar{u} \, \varphi(b(\bar{v}, \bar{u})) \rightarrow p(\bar{v})$$

where $\forall \bar{v} = \forall v_1, \ldots, \forall v_n$ and $\exists \bar{u} = \exists u_1, \ldots, \exists u_w$, like done in logic programs [14].

## 4 Experiments

In this section we analyze the following research questions: **Generalization**—Can standard/relational CBMs generalize well in relational tasks? Can standard/relational CBMs generalize in out-of-distribution settings? [3]

**Interpretability**—Can relational CBMs provide meaningful explanations for their predictions? Are concept/rule interventions effective in relational CBMs? **Efficiency**—Can relational CBMs generalize in low-data regimes? Can relational CBMs correctly predict concept/task labels with scarce concept train labels?

**Data & task setup.** We investigate our research questions using 7 relational datasets on image classification, link prediction and node classification. We introduce two simple but not trivial relational benchmarks, namely the Tower of Hanoi and Rock-Paper-Scissors (RPS), to demonstrate that standard CBMs cannot even solve very simple relational problems. The Tower of Hanoi is composed of 1000 images of disks positioned at different heights of a tower. Concepts include whether disk $i$ is larger than $j$ (or vice versa) and whether disk $i$ is directly on top of disk $j$ (or vice versa). The task is to predict for each disk whether it is well-positioned or not. The RPS dataset is composed of 200 images showing the characteristic hand-signs. Concepts indicate the object played by each player and the task is to predict whether a player wins, loses, or draws. We also evaluate our methods on real-world benchmark datasets specifically designed for relational learning: Cora, Citeseer, [30], PubMed [23] and Countries on two increasingly difficult splits [28]. Additional details can be found in App. A.1 and App. A.5.

**Models.** We compare R-CBMs against state-of-the-art concept bottleneck architectures, including CBMs with linear and non-linear task predictors (**CBM-Linear** and **CBM-Deep**) [13], a flat version (**Flat-CBM**) where each prediction is computed as a function of the full set of ground atoms, but also with **Feedforward** and **Relational black-box** architectures. We also compared against DeepStochLog [33], a state-of-the-art NeSy system, and other KGE specific models for the studied KGE tasks. Our relational models include an R-CBM with DCR predictor (**R-DCR**) and its direct variant, using only 5 supervised examples per-predicate (**R-DCR-Low**). We also considered a non-interpretable R-CBM version where the predictions are based on an unrestricted predictor processing the atom representations (**R-CBM-Emb**). In the experiments, the loss function was selected to be the

---

[3]The code to replicate the experiments presented in this paper is available at `https://github.com/diligmic/RCBM-Neurips2024`.

Table 1: **Models' performance on task generalization**. R-CBMs generalize well in relational tasks. △ indicates methods that cannot be applied due to the dataset structure. OOT indicates out-of-time training due to large domains.

| MODEL | | FEATURES | | | DATASETS | | | | | | |
|---|---|---|---|---|---|---|---|---|---|---|---|
| Class | Name | Rel. | Interpr. | Rules | RPS (ROC-AUC ↑) | Hanoi (ROC-AUC ↑) | Cora (Accuracy ↑) | Citeseer (Accuracy ↑) | PubMed (Accuracy ↑) | Countries S1 (MRR ↑) | Countries S2 (MRR ↑) |
| Black Box | Feedforward | No | No | No | $64.46 \pm 0.63$ | $54.36 \pm 0.25$ | $46.86 \pm 2.94$ | $45.15 \pm 3.79$ | $68.83 \pm 0.85$ | △ | △ |
| | Relational | Yes | No | No | $100.00 \pm 0.00$ | $98.77 \pm 0.60$ | $76.66 \pm 1.34$ | $\mathbf{68.32 \pm 0.71}$ | $74.93 \pm 0.30$ | $91.56 \pm 1.02$ | $87.87 \pm 0.64$ |
| | Relational + C&S | Yes | Yes | No | – | – | $63.59 \pm 0.95$ | $64.89 \pm 4.01$ | $78.64 \pm 1.42$ | △ | △ |
| NeSy | DeepStochLog | Yes | Yes | Given | $100.00 \pm 0.00$ | $100.00 \pm 0.00$ | $77.52 \pm 0.58$ | $67.03 \pm 0.97$ | $74.88 \pm 1.24$ | △ | △ |
| CBM | CBM-Linear | No | Yes | No | $54.74 \pm 2.50$ | $51.02 \pm 0.14$ | △ | △ | △ | △ | △ |
| | CBM-Deep | No | Partial | No | $53.01 \pm 1.59$ | $54.94 \pm 0.28$ | △ | △ | △ | △ | △ |
| | DCR | No | Yes | Learnt | $64.48 \pm 0.64$ | $54.58 \pm 0.25$ | △ | △ | △ | △ | △ |
| R-CBM (Ours) | R-CBM-Linear | Yes | Yes | No | $51.04 \pm 1.99$ | $100.00 \pm 0.00$ | $76.37 \pm 1.80$ | $67.16 \pm 2.05$ | $64.46 \pm 9.53$ | $93.81 \pm 2.42$ | $\mathbf{92.27 \pm 2.84}$ |
| | R-CBM-Deep | Yes | Partial | No | $100.00 \pm 0.00$ | $100.00 \pm 0.00$ | $\mathbf{78.42 \pm 1.48}$ | $66.92 \pm 0.75$ | $75.36 \pm 1.36$ | $92.75 \pm 2.12$ | $91.81 \pm 2.01$ |
| | Flat-CBM | Yes | Yes | No | $50.74 \pm 0.54$ | $82.91 \pm 5.82$ | OOT | OOT | OOT | OOT | OOT |
| | R-DCR | Yes | Yes | Learnt | $98.77 \pm 0.31$ | $99.99 \pm 0.01$ | $78.30 \pm 2.10$ | $66.84 \pm 1.52$ | $\mathbf{75.86 \pm 1.74}$ | $\mathbf{98.33 \pm 2.05}$ | $92.19 \pm 1.52$ |
| | R-DCR-Low | Yes | Yes | Learnt | $98.11 \pm 1.09$ | $90.62 \pm 2.97$ | △ | △ | △ | △ | △ |

Table 2: MRR and Hits@N metrics on the test set of the WN18RR and FB15k-237dataset. The competitor results have been taken from Cheng et al. [2] or from the original datasets.

| Class | Name | WN18RR | | | FB15k-237 | | |
|---|---|---|---|---|---|---|---|
| | | (MRR ↑) | (Hits@1 ↑) | (Hits@10 ↑) | (MRR ↑) | (Hits@1 ↑) | (Hits@10 ↑) |
| Black Box | DistMult | 0.42 | 0.382 | 0.507 | 0.24 | 0.155 | 0.419 |
| | ConvE | 0.43 | 0.401 | 0.525 | 0.33 | 0.237 | 0.501 |
| | ComplEx | 0.44 | 0.410 | 0.512 | 0.26 | 0.163 | 0.452 |
| | ComplEx-N3 | 0.48 | - | **0.570** | **0.37** | - | **0.560** |
| Logic Based | NLIL | 0.30 | 0.201 | 0.335 | 0.25 | - | 0.324 |
| | RNNLogic with emb. | 0.48 | 0.446 | 0.558 | 0.34 | 0.252 | 0.530 |
| | RLogic | 0.47 | 0.443 | 0.537 | 0.31 | 0.203 | 0.501 |
| | LPRules | 0.46 | 0.422 | 0.532 | 0.26 | 0.170 | 0.402 |
| | LatentLogic | 0.48 | **0.497** | 0.553 | 0.32 | 0.212 | 0.514 |
| R-CBM (ours) | R-CBM-Emb | **0.49** | 0.447 | 0.559 | 0.35 | 0.254 | 0.531 |
| | R-DCR | 0.47 | 0.419 | 0.563 | 0.35 | **0.255** | 0.533 |

standard cross-entropy loss. Further details are in App. A.2. **Evaluation.** We measure generalization using standard metrics, i.e., Area Under the ROC curve [9] for multi-class classification, accuracy for binary classification, and Mean Reciprocal Rank (MRR) for link prediction, MRR and Hits@N for KGE tasks. We use these metrics to measure generalization across all experiments, including out-of-distribution scenarios, low-data regimes, and interventions. We report additional experiments and further details in App. A.3.

## 5 Key Findings

### 5.1 Generalization

**Standard CBMs do not generalize in relational tasks (Table 1).** Standard CBM best task performance $\sim 55\%$ ROC-AUC is just above a random baseline. This result directly stems from the architecture of existing CBMs, which can process only one input entity at a time. The experiments validate that this design fails on relational tasks that inherently involve multiple entities. Naive attempts to address the relational setting, like Flat-CBMs, lead to a significant drop in task generalization performance ($-17\%$ in Hanoi), and become intractable when applied to larger datasets (e.g., Cora, Citeseer, PubMed, Countries). In RPS, instead, Flat-CBMs performance is close to random as the linear predictor for this model can not well approximate the required non-linear combination of concepts. These findings expose the limitations of existing CBMs when applied to relational tasks and justify the need for relational CBMs.

**R-CBMs generalize well in relational tasks (Tables 1 and 2.** Relational concept bottleneck models match the generalization performance of relational black-box models (GNNs and KGEs) in relational tasks. For example, R-CBMs exhibit gains of up to $7\%$ MRR (Countries S1), and at most a $1\%$ loss in accuracy (Citeseer) w.r.t. relational black-boxes. In larger KGEs like WN18RR , R-DCR beats standard KGEs and is competitive against state-of-the-art custom logic-based solutions, while being more general. Non-interpretable solutions R-CBM-Emb are admitted by our relational formulation when $f_c$ is selected to be a generic MLP blackbox. Since R-CBM-Emb and R-DCR only differ for the selection of the predictor (interpretable for R-DCR), a comparison of the results of these models on WN18RR provides a direct measurement of the performance decay due to the additional interpretability. Relational CBMs employing a simple linear layer as task predictor

Table 3: **CBMs response to interventions**. R-CBMs effectively respond to human interventions.

| | RPS | | Hanoi | |
| --- | --- | --- | --- | --- |
| | **Before Interv.** | **After Interv.** | **Before Interv.** | **After Interv.** |
| R-CBM-Linear | $49.46 \pm 1.11$ | $47.83 \pm 2.19$ | $49.26 \pm 1.01$ | $\mathbf{100.00 \pm 0.00}$ |
| R-CBM-Deep | $51.35 \pm 2.00$ | $82.02 \pm 6.34$ | $50.07 \pm 0.74$ | $\mathbf{100.00 \pm 0.00}$ |
| R-DCR | $\mathbf{54.47 \pm 1.64}$ | $\mathbf{100.00 \pm 0.00}$ | $49.48 \pm 0.35$ | $\mathbf{100.00 \pm 0.00}$ |
| CBM-Linear | $49.41 \pm 0.89$ | $47.75 \pm 1.81$ | $50.01 \pm 0.16$ | $55.03 \pm 0.53$ |
| CBM-Deep | $50.83 \pm 0.93$ | $47.78 \pm 1.94$ | $\mathbf{50.44 \pm 0.46}$ | $60.14 \pm 0.46$ |
| DCR | $51.22 \pm 1.26$ | $49.07 \pm 1.60$ | $50.00 \pm 0.00$ | $50.00 \pm 0.00$ |

(R-CBM-Linear) underfit tasks demanding on non-linear combinations of concepts (e.g., RPS). In such scenarios, a deeper task predictor (e.g., R-CBMs Deep) trivially solves the issue, but it also hampers interpretability. R-DCRs address this limit providing accurate predictions while generating high-quality rule-based explanations (Table 4). It also matches generalization performance of neural symbolic system DeepStochLog [33], which is provided with ground truth rules.

**R-CBMs generalize in out-of-distribution settings where the number of entities changes at test time (Figure 4).** R-CBMs show robust generalization performances even in out-of-distribution conditions where the number of entities varies between training and testing. To assess generalization in these extreme conditions, we use the Tower of Hanoi dataset, where test sets of increasing complexity are generated by augmenting the number of disks in a tower. We observe that a naive approach, such as Flat-CBMs, immediately breaks as soon as we introduce a new disk in a tower, as its architecture is designed for a fixed number of input entities. In contrast, R-CBMs are more resilient, as we observe a smooth performance decline from $\sim 100\%$ ROC-AUC (with 3 disks in both training and test sets) to around $\sim 85\%$ in the most challenging conditions (with 3 disks in the training set and 7 in the test set).

## 5.2 Interpretability

**R-CBMs support effective interventions (Table 3).**

CBM architectures allow human interaction with the learnt concepts to intervene on mispredicted concepts during testing to improve the final predictions. In our experiments we assess CBMs' response to interventions on the RPS and Hanoi datasets. We set up the evaluation by generating a batch of adversarial test samples that prompt concept encoders to mispredict $\sim 50\%$ of concept labels by introducing a strong random noise in the input features drawn from the uniform distribution $\mathcal{U}(0, 20)$. In our findings, we note that R-CBMs positively respond to test-time concept interventions by increasing their task perfor-

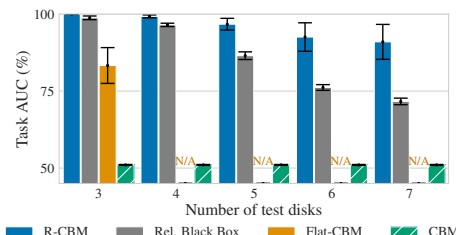

Figure 4: **Model generalization on Hanoi OOD on the number of disks.** Only R-CBMs are able to generalize effectively to settings larger than the ones they are trained on.

mance. This contrasts with standard CBMs, where perfect concept predictions are not enough to solve the relational task. Notably, the RPS dataset poses a significant challenge for relational CBMs equipped with linear task predictors, as the task depends on a non-linear combination of concepts. Expanding our investigation to DCRs, we expose another dimension of human-model interaction: *rule interventions*. Applying both concept and rule interventions, we observe that R-DCRs perfectly predict all adversarial test samples.

**Relational Concept Reasoners discover semantically meaningful rules (Table 4).** Among CBMs, a key advantage of DCRs lies in the dual role of generating rules which serve for both generating and explaining task predictions. Table 4 shows instances of R-DCR explanations, confirming that R-DCR discovers rules aligned with known ground truths across diverse datasets (e.g., $wins(X) \leftarrow \neg rock(X) \wedge paper(X) \wedge \neg scissors(X) \wedge rock(Y) \wedge \neg paper(Y) \wedge \neg scissors(Y)$ in RPS). Notably, R-DCR discovers meaningful rules even in low data regimes (R-DCR-Low) and when the correct rules are unknown, such as in Cora, Citeseer and PubMed.

Table 4: **Rules extracted by relational DCRs**. In Hanoi, we remove negative atoms for brevity.

| Dataset | Examples of learnt rules |
|---|---|
| RPS | $\forall v, \exists u.\ wins(v) \leftarrow \neg rock(v) \wedge paper(v) \wedge \neg scissors(v) \wedge rock(u) \wedge \neg paper(u) \wedge \neg scissors(u)$ |
| | $\forall v, \exists u.\ loses(v) \leftarrow \neg rock(v) \wedge \neg paper(v) \wedge scissors(v) \wedge rock(u) \wedge \neg paper(u) \wedge \neg scissors(u)$ |
| | $\forall v, \exists u.\ ties(v) \leftarrow rock(v) \wedge \neg paper(v) \wedge \neg scissors(v) \wedge rock(u) \wedge \neg paper(u) \wedge \neg scissors(u)$ |
| Hanoi | $\forall v, \exists u_1, u_2.\ correct(v) \leftarrow top(u_1, v) \wedge top(v, u_2) \wedge larger(v, u_1) \wedge larger(u_2, v) \wedge larger(u_2, u_1)$ |
| | $\forall v, \exists u_1, u_2.\ correct(v) \leftarrow top(v, u_2) \wedge top(u_1, u_2) \wedge top(u_2, u_1) \wedge larger(v, u_2) \wedge larger(u_2, v)$ |
| Cora | $\forall v, \exists u.\ nn(v) \leftarrow nn(u) \wedge \neg rl(u) \wedge \neg rule(u) \wedge \neg probModels(u) \wedge \neg theoru(u) \wedge \neg gene(u) \wedge cite(v, u)$ |
| PubMed | $\forall v, \exists u.\ type1(v) \leftarrow type1(u) \wedge \neg type2(u) \wedge \neg experimental(u) \wedge cite(v, u)$ |
| Countries | $\forall v_1, v_2, \exists u.\ locatedIn(v_1, v_2) \leftarrow locatedIn(v_1, u) \wedge locatedIn(u, v_2)$ |

Table 5: **Data efficiency** (Citeseer dataset). Relational CBMs are more robust than an equivalent relational black-box when reducing the amount of supervised training nodes.

| % Supervision | 100% | 75% | 50% | 25% |
|---|---|---|---|---|
| Rel. Black-Box | **68.32 ± 0.71** | 66.02 ± 0.67 | 46.46 ± 2.01 | 7.70 ± 0.0 |
| R-CBM-Linear | 67.16 ± 2.05 | 65.96 ± 0.87 | **57.07 ± 3.74** | **16.92 ± 4.83** |
| R-CBM-Deep | 66.92 ± 0.75 | 64.08 ± 1.99 | 56.59 ± 1.05 | 12.25 ± 3.53 |
| R-DCR | 66.89 ± 1.52 | **66.42 ± 1.66** | 52.30 ± 3.15 | 16.52 ± 1.29 |

## 5.3 Low data regimes

**R-CBMs generalize better than relational black-boxes in low-data regimes (Table 5).** The ability of relational CBMs and relational black box models was compared on the Citeseer dataset as the number of labeled nodes decreased to 75%, 50%, and 25%. While no significant difference was observed with ample training data, a growing advantage for relational CBMs over relational black box models emerged in scenarios of scarce data. The intermediate predictions related to incoming atoms likely have a crucial regularization effect, particularly in scenarios with limited data.

**R-DCR accurately makes interpretable predictions with very few atom supervisions (Table 1).** R-DCR-Low is able to learn an interpretable relational predictor when reducing the training data to 5 labeled atoms for each predicate. Indeed, the supervisions are crucial to establish an alignment between human knowledge and the model on the semantics of logical explanations. The alignment can be perfectly achieved in RPS, where the predictions are mutually exclusive. On the Hanoi dataset, learning the relational binary concepts $larger$ and $top$ from 5 examples is challenging, leading to slightly decreased overall performance.

## 6 Discussion

**Related work on CBMs.** Concept bottleneck models [13] inspired several works focusing on improved generalization [15, 5, 31], explanations [3, 1] and robustness [17, 10, 36, 11]. Despite these efforts, the application of CBMs to relational domains remains unexplored. Filling this gap, our framework allows relational CBMs to (i) effectively solve relational tasks, and (ii) generalize the explanatory capabilities of these models from propositional to relational.

**Related work on GNNs.** R-CBMs and relational black-boxes (such as GNNs) share similarities in considering the relationships between multiple entities when solving a given task. The prediction computation of R-CBMs is based on a message passing paradigm which is similar to message-passing in graph neural networks [7], which is a special case of the proposed architecture. However, relational CBMs can also define aggregations based on a semantically meaningful concepts, allowing the extraction of explanations, which can not be done by GNNs.

**Related work on ILP.** Among our considered R-CBMs, R-DCR is the only one learning a set of logic rules, hence we can draw some parallels with some algorithmic solutions defined in Inductive Logic Programming (ILP) [21]. R-DCR bottlenecks are connected to ILP mode declarations or metarules [22], which define the search space. However, while ILP involves searching through a hypothesis space of possible logical rules, guided by principles like consistency, coverage, and simplicity, R-DCR is different as it searches this space via gradient descent over a continuous relaxation of the logic, and the logic formulas are learnt by exploiting neural architectures.

**Related work on Neuro-Symbolic AI.** Neural Theorem Provers (NTP) [27] and their more scalable variations [20] combine formal proof proving with neural networks for efficiency via a trainable

heuristic search. However, R-CBMs are more general than NTP –or other specific rule learners–indeed their interpretability is not restricted to rule learning, but rather rely on concept interventions. Moreover, unlike NTP, when dealing with logic rules the R-CBM framework is not limited to Horn clauses, and it can be applied in classic CBMs setups where inputs are not symbolic, but images. Moreover, R-CBMs' templates can represent all the rules using a (subset of a) specified list of atoms in the body at the same time, and the embeddings will be used to determine which actual rule to instantiate in each given context (like in R-DCR). On the other hand, NTP's approach to rule learning is to enumerate all possible rules and let the learning decide which rules are useful. Please note that this approach is not scalable to larger KGs because of the combinatorial explosion of the number of rules when there are many predicates in the dataset.

**Limitations.** A limitation of relational CBMs consists in their limited scalability to very large domains. This limitation is shared with all existing relational systems, and most relational models have to rely on simplifying heuristics to scale to large relational structures like knowledge graphs [37, 26]. Another limitation of R-CBMs lays in the need for the definition of a relational concept bottlenecks, which acts as an architectural inductive bias, restricting the search space. Future extensions of relational CBMs can relax the need of an external template definition by including an automatic calibration of template widths, the construction of reduced set of variables' substitutions, or the automatic generation of the relational templates.

**Conclusions.** This work presents R-CBMs, a family of concept bottleneck models designed for relational tasks. The results of our experiments show that R-CBMs: (i) match the generalization performance of existing relational black-boxes, (ii) support the generation of quantified concept-based explanations, (iii) effectively respond to test-time interventions, and (iv) withstand demanding settings including out-of-distribution scenarios, and low data regimes. R-CBMs represent a significant extension of standard CBMs, and pave the way to further investigations using CBMs to improve interpretability in GNNs and to explain KGE predictions.

## Acknowledgments and Disclosure of Funding

PB acknowledges support from SNSF project TRUST-ME (No. 205121L_214991). This research has also received funding from the KU Leuven Research Fund (STG/22/021, CELSA/24/008) and from the Flemish Government under the "Onderzoeksprogramma Artificiële Intelligentie (AI) Vlaanderen" programme. FG has been supported by the Partnership Extended PE00000013 - "FAIR - Future Artificial Intelligence Research" - Spoke 1 "Human-centered AI". MD was supported by TAILOR and HumanE-AI-Net, projects funded by EU Horizon 2020 research and innovation programme under GA No 952215 and No 952026, respectively. This project has also be partially supported by the EU Framework Program for Research and Innovation Horizon under the Grant Agreement No 101073307 (MSCA-DN LeMuR).

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

# A  Appendix

## A.1  Datasets

### A.1.1  Rock-Paper-Scissors

We build the Rock-Paper-Scissors (RPS) dataset by downloading images from Kaggle: `https://www.kaggle.com/datasets/drgfreeman/rockpaperscissors?resource=download`. The dataset contains images representing the characteristic hand-signs annotated with the usual labels "rock", "paper", and "scissors". To build a relational dataset we randomly select 200 pairs of images and defined the labels wins/ties/loses according to the standard game-play. To train the models we select an embedding size of 10.

### A.1.2  Tower of Hanoi

We build the Tower of Hanoi (Hanoi) dataset by generating disk images with matplotlib. We randomly generate 1000 images representing disks of different sizes in $[1, 10]$ and at different heights of the tower in $[1, 10]$. We annotate the concepts $top(u, v), larger(u, v)$ using pairs of disks according to the usual definitions. We define the task label of each disk according to whether the disk is well positioned following the usual definition that a disk is well positioned if the disk below (if any) is larger, and the disk above (if any) is smaller. To train the models we select an embedding size of 50.

### A.1.3  Cora, Citeseer, PubMed, Countries

For the experiments in Table 1, we exploit the standard splits of the Planetoid Cora, Citeseer and PubMed citation networks, as defined in Pytorch Geometric `https://pytorch-geometric.readthedocs.io/en/latest/modules/datasets.html`. The classes of documents are used both for tasks and concepts

The Countries dataset (ODbL licence) [4] defines a set of countries, regions and sub-regions as basic entities. We used splits and setup from Rocktaschel et al. [28], which reports the basic statistics of the dataset and also defines the tasks S1, S2 used in this paper.

## A.2  Baselines

### A.2.1  Exploiting prior knowledge

Additionally, we can use prior knowledge to optimize the template and the aggregation by excluding concept atoms in $b(\bar{v}, \bar{u})$ and groundings in $\Theta$ that are not relevant to predict the task. This last simplification is crucial anytime we want to impose a *locality* bias, and it is also at the base of the heuristics that are commonly used in extension of knoweldge graph embeddings with additional knowledge [26, 37, 4].

### A.2.2  Cora, Citeseer, PubMed

Slash notation $a/b/c$ indicates parameters for $cora/citeseer/pubmed$ when different.

R-CBMs exploit the same concept encoder $g_i$, which corresponds to an MLP with 2 hidden layers of size $32/16/16$ followed by an output layer of size $6/7/3$ classes. Activation functions are LeakyReLu. The blackbox **feedforward** network is equivalent to the one of the CBM models. The blackbox **relational** model is a GCN with 2 layers of size 16. Node features for R-CBM models are initialized with the last embeddings of the GCN. **R-CMB Deep** task predictor exploits a 2 layer MLP with 1 hidden layers of size $32/16/16$ followed by an output layer of size 1. Activation functions are LeakyReLu. **R-DCR** exploits, as $filter$ and $sign$ functions a linear layer of size $32/16/16$. **DeepStochLog** exploits the same concept encoder as neural predicate. It exploits also the pretraining using a GCN. As task predictor, it exploits a SDCG grammar implementing the rule $cite(v_1, v_2) \to class_i(v_1) \iff class_i(v_2)$.

---

[4] `https://github.com/mledoze/countries`

### A.2.3 Countries

The DistMult Knowledge Graph Embeddings (KGE) [34] was used as BlackBox relational baseline for the Countries S1 and S2 datasets. We varied the embedding sizes in the set $\{10, 20, 50, 100, 200, 300\}$ and selected the best results on the validation set. The DistMult KGE was used as a basic concept encoder for CMBs. The R-CBM-Linear computes the concepts via linear layer followed by the KGE output layer. The R-CBM-Deep computes the concepts via an MLP with 2 hidden layers followed by a KGE output layer. Activation functions are ReLu.

### A.3 Experimental Details and Additional experiments

All experiments have been carried out on a machine with a Intel i7 CPU, 128GB RAM. Running times for all experiments are within 1 hour, with the exception of the link prediction experiment on WN18RR, which took 14h:20m.

### A.3.1 Training Hyperparameters

In all synthetic tasks, we generate datasets with 3,000 samples and use a traditional 70%-10%-20% random split for training, validation, and testing datasets, respectively. During training, we then set the weight of the concept loss to $\lambda = 0.1$ across all models. We then train all models for 3000 epochs using full batching and a default Adam [12] optimizer with learning rate $10^{-4}$. KGE experiments have used the Complex and Rotate KGE encoder and scorer function for $g_r, s$ in the Countries and WN18RR datasets, respectively.

### A.3.2 Data Efficiency

As explained in Section A.2.2, the relational CBMs exploits the features obtained by pretraining on a GNN on the same data split. Such pretraining is beneficial only in high-data settings (i.e. 100%, 75% and 50%). On low data regime (i.e. 25%), pretrained features are worse than original features. In these cases, we train the different baselines from scratch by using directly the low level features of the documents.

For training R-CBM models on the WN18RR dataset, we used the heuristic, commonly employed in the NeSy community, of instantiating only hyperedges where all source atoms are observed in the training set [37].

### A.3.3 Countries

The task consists of predicting the unknown locations of a country, given the evidence in form of country neighbourhoods and some known country/region locations. The entities are divided into the $C, R, W$ domains referring to the countries, regions and continents, respectively. The predicate $locIn(v_1, v_2)$ determines the location of a country in a region or continent, with the variables $(v_1, v_2) \in C \times R \cup W$ or $(v_1, v_2) \in R \times W$. The country neighbourhoods are determined by the predicate $neighOf(v_1, v_3)$ with the variables $v_1, v_3 \in C$.

The entities in the dataset are a set of countries, regions and continents represented by their corresponding feature vectors as computed by a DistMult KGE (see baselines). The concept datasets are respectively the set $D_{c_{locIn}} = (C \times R) \cup (R \times W)$ and $D_{c_{neighOf}} = C \times C$. The task dataset $D_{y_{locIn}} = C \times W$ is formed by queries about the location of some countries within a continent. The templetized relational concept bottleneck is defined as:

$$\mathcal{N}(locIn(v_1, v_2)) = [locIn(v_1, u_1), locIn(u_1, v_2), neighOf(v_1, u_2), locIn(u_2, v_2)] .$$

Finally, the cross entropy loss was used both for functions for concepts.

### A.3.4 Hard-to-classify samples

R-CBMs can also be used to find easy/hard samples, similar to what done for standard CBMs in [6]. In our framework, we consider as hard examples the ones whose prediction is highly uncertain when using a CBM with a propositional template (see CBM-Deep rows in Table 6. When using a relational template, instead, we verified that the distribution of the prediction uncertainty significantly decreases. We show this in Figure 5, where the prediction uncertainty decreases when transitioning

from a propositional to a relational template. Table 6 shows the concept/task activation for the hardest example to classify using the propositional template (high uncertainty) and the corresponding predictions when using the relational template (low uncertainty).

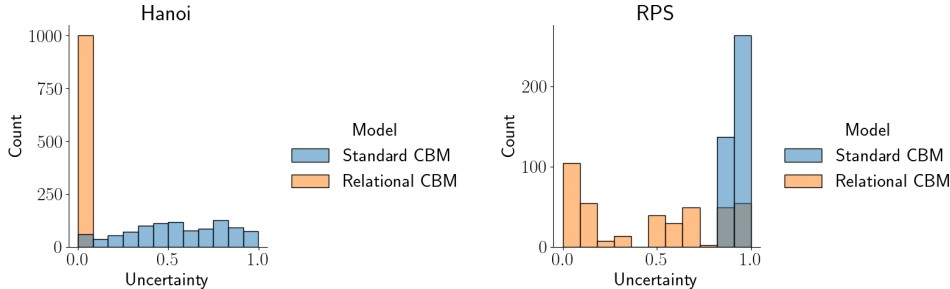

Figure 5: Distribution of class prediction uncertainty comparing CBMs using relational vs propositional bottlenecks.

Table 6: Examples of hard to classify examples using a propositional bottleneck (used by CBM-Deep) which become easy to classify when using a relational bottleneck (used by R-CBM-Deep).

| Dataset | Model | Concept activations | Task activations |
|---|---|---|---|
| **RPS** | CBM-Deep | rock(X) = .00, paper(X) = 1.0, scissors(X) = .00 | wins(X) = .31, ties(X) = .32, loses(X) = .36 |
| | R-CBM-Deep | rock(X) = .00, paper(X) = 1.0, scissors(X) = .00, rock(Y) = .01, paper(Y) = .00, scissors(Y) = .96 | wins(X) = .01, ties(X) = .01, loses(X) = 1.0 |
| **Hanoi** | CBM-Deep | Position0(X) = .00, Position1(X) = .00, Position2(X) = .02, Position3(X) = .65, Position4(X) = .35, Position5(X) = .00, Position6(X) = .00, Size0(X) = .00, Size1(X) = .20, Size2(X) = .47, Size3(X) = .19, Size4(X) = .01, Size5(X) = .00, Size6(X) = .00, Size7(X) = .00, Size8(X) = .00, Size9(X) = .00 | correct(X) = .50 |
| | R-CBM-Deep | top(X,Y) = 1.0, top(Y,X) = .00, top(X,Z) = .00, top(Z,X) = .00, top(Y,Z) = 1.0, top(Z,Y) = .00, larger(X,Y) = .00, larger(Y,X) = 1.0, larger(X,Z) = .00, larger(Z,X) = 1.0, larger(Y,Z) = .00, larger(Z,Y) = 1.0 | correct(X) = 1.0 |

### A.3.5 Completeness scores

While Table 5 reported an evaluation of concept efficiency, here we report the completeness scores of each concept-based model wrt the relational baseline, following Equation 1 in [35]. The results are shown in Table 7.

| | RPS | Hanoi | Cora | Citeseer | PubMed | Countries S1 | Countries S1 |
|---|---|---|---|---|---|---|---|
| CBM-Linear | 32.44 | 2.09 | N/A | N/A | N/A | N/A | N/A |
| CBM-Deep | 29.86 | 10.12 | N/A | N/A | N/A | N/A | N/A |
| DCR | 46.98 | 9.39 | N/A | N/A | N/A | N/A | N/A |
| R-CBM Linear | 26.92 | 102.52 | 98.91 | 93.66 | 58.00 | 105.41 | 111.61 |
| R-CBM Deep | 100.00 | 102.52 | 106.60 | 92.35 | 101.72 | 102.86 | 110.40 |
| R-DCR | 98.16 | 102.50 | 106.15 | 91.92 | 103.73 | 116.28 | 111.40 |

Table 7: Completeness scores of each concept-based model wrt the relational black-box baseline.

### A.4 Relational Task Predictors

In standard CBMs, a wide variety of task predictors $f$ have been proposed on top of the concept encoder $g$, defining different trade-offs between model accuracy and interpretability. In the following, we resume how we adapted a selection of representative models for $f$ to be applicable in a relational setting (fixing for simplicity $o = 0$). These are the models that we will compare in the experiments (Section 4 and 5).

**Relational Concept Bottleneck Model Linear (R-CBM-Linear)** The most basic task predictor employed in standard CBMs is represented by a single linear layer [13]. This choice guarantees a high-degree of interpretability, but may lack expressive power and may significantly underperform whenever the task depends on a non-linear combination of concepts. In the relational context, we define it as following:

$$f(\theta_{\bar{u}} b(\bar{x}, \bar{u})) = W\theta_{\bar{u}} b(\bar{x}, \bar{u}) + w_0 \tag{2}$$

**Deep Relational Concept Bottleneck Model (Deep R-CBM)** To solve the linearity issue of R-CBM, one can increase the number of layers employed by the task predictor (as also proposed in [13]). In the relational context we can define a Deep R-CBM as following:

$$\text{Deep R-CBM:} \quad f(\theta_{\bar{u}} b(\bar{x}, \bar{u})) = \varphi(\theta_{\bar{u}} b(\bar{x}, \bar{u})), \tag{3}$$

where we indicate with $MLP$ a multi-layer perceptron. However, the interpretability between concept and task predictions is lost, since MLPs are not transparent. Further, the ability of a Deep R-CBM to make accurate predictions is totally depending on the existence of concepts that univocally represent the tasks, hence being possibly very inefficacy.

**Relational Deep Concept Reasoning (R-DCR)** [5] proposed to encode concepts by employing concept embeddings (instead of just concept scores), improving CBMs generalization capabilities, but affecting their interpretability. Then [1] proposed to use these concept embeddings to generate a symbolic rule which is then executed on the concept scores, providing a completely interpretable prediction. We adapt this model in the relational setting:

$$\text{R-DCR:} \quad f(\theta_{\bar{u}} b(\bar{x}, \bar{u})) = \varphi(\theta_{\bar{u}} b(\bar{x}, \bar{u})), \tag{4}$$

where $\varphi$ indicates the rule generated by a neural module working on the concept embeddings. For further details on how $\varphi$ is learned, please refer to [1]. Since the logical operations in R-DCR are governed by a semantics specified by a t-norm fuzzy logic [8], whenever we use this model we require the aggregation operation $\oplus$ used in Eq. 3.2 to correspond to a fuzzy OR. The max operator corresponds to the OR within the Gödel fuzzy logic.

**Relational Deep Embedding Reasoning (R-CBM-Emb)** [18] proposes a latent relational process, which computes the atom representations using the presentations of other atoms that co-occur in the same ground formula. The final readout is based on an MLP processing the final atom representation. This model can exploit the rich reletional representations developed as atom embeddings, but it acts as a blackbox in terms of explanations of how the decision process takes form. This model can be implemented in our general model structure by restricting the $f_c$ function to only process the $h_c^t$ embeddings as input, such that:

$$
\begin{aligned}
h_c^t(A) &= u_{r(l)} \left( h^{t-1}(A), \left[ h^{t-1}(B) \right]_{B \in \mathcal{N}_c(A)} \right) \\
y_c^t(A) &= MLP \left( h_c^t(A) \right).
\end{aligned}
$$

**R-DCR-Low** R-DCR-Low is a version of R-DCR that is trained by providing the atom supervisions for only 5 incoming hyperedges. Its architecture and learning is entirely identical to DCR except for two variants:

- Since DCR strongly depends on crisp concepts prediction for learning good and interpretable rule, in absence of sufficient supervision, we need a different way to obtain crisp predictions. To this end we substitute the standard sigmoid and softmax activation functions for concept predictors $g_i$ with discrete differentiable sample from a bernoulli or categorial distributions. The differentiability is obtained by using the Straight Through estimators provided by PyTorch.
- Since the backward signal from DCR can be very noisy at the beginning of the learning, we add a parallel task predictor (and a corresponding loss term), completely identical to the one of a R-CBM-Deep model. Such predictor only guides the learning of the concepts during training by a cleaner backward signal but is discarded during test, leaving a standard DCR architecture.

**Flat Concept Bottleneck Model (Flat-CBM)** assumes each prediction to be computed as a function of the full set of ground atoms. This model has limited scalability but it is introduced for comparison reasons in the experimental section.

### A.5 Code, Licences, Resources

**Libraries** For our experiments, we implemented all baselines and methods in Python 3.7 and relied upon open-source libraries such as PyTorch 1.11 [24] (BSD license) and Scikit-learn [25] (BSD license). To produce the plots seen in this paper, we made use of Matplotlib 3.5 (BSD license). We will release all of the code required to recreate our experiments in an MIT-licensed public repository.

**Resources** All of our experiments were run on a private machine with 8 Intel(R) Xeon(R) Gold 5218 CPUs (2.30GHz), 64GB of RAM, and 2 Quadro RTX 8000 Nvidia GPUs. We estimate that approximately 50-GPU hours were required to complete all of our experiments.

### Ethical Statement

There are no ethical issues.

