# OpenReview forum: "Relational Concept Bottleneck Models"
_NeurIPS.cc/2024/Conference — NeurIPS 2024 poster_

### Official Review · Reviewer_XAga · 2024-06-13

**Soundness:** 3
**Presentation:** 4
**Contribution:** 3
**Rating:** 7
**Confidence:** 3

**Summary:**

The authors propose Relational Concept Bottleneck Models, a family of relational deep learning methods that utilize concept bottleneck models to provide interpretable task predictions. R-CBMs are shown to predict well in various settings, matching the performanace of black-box models.

**Strengths:**

The paper studies an important problem

The paper is clearly written and experimental results are easy to follow

The results show strong and consistent improvements

**Weaknesses:**

It might be nice to include a comparison to a simpler graph-based method that is not black box, e.g. [C&S](https://arxiv.org/abs/2010.13993) and a discussion of how the choice of relational task predictor and aggregation affect interpretability

Would be interesting to see how R-CBM compares with CBM on datasets that aren't specifically designed for relational prediction.

**Questions:**

What is the explanation for why R-CBM-Linear does not effectively respond to interventions for RPS?

**Limitations:**

Adequately addressed

---

> ### Author Rebuttal · Authors · 2024-08-06
>
> *R-CBM vs CBM comparison on non-relational datasets:*
> **Vanilla CBMs are special cases of R-CBMs in non-relational domains (as described in L158-167)**, hence R-CBMs’ results on non-relational datasets (where predicates are unary) would be identical to propositional CBMs’ results (architecture and losses are both identical).
>
> *Why does R-CBM-Linear not effectively respond to interventions for RPS?*
> **The RPS task is a non-linear combination of concepts, so it can’t be solved with a linear model such as R-CBM-Linear.** When using non-linear task predictors (R-DCR or R-CBM-Deep), the task can be solved, and interventions become effective.
>
> *Comparison with a simpler graph-based method that is not black box:*
> **We thank the reviewer for the suggestion. We have added the comparison with the method Correct and Smooth (C&S) as suggested (cf. Table A of attached PDF).** Please, see also the common answer on baselines for more details. We added this baseline in Table 1 in the revised version of our paper.

---

> > ### Comment · Reviewer_XAga · 2024-08-08
> > **Thanks for your comments**
> >
> > I thank the authors for their response, but maintain my rating of 7

---

### Official Review · Reviewer_raJm · 2024-07-01

**Soundness:** 3
**Presentation:** 3
**Contribution:** 3
**Rating:** 5
**Confidence:** 4

**Summary:**

This paper proposed a Relational Concept Bottleneck Models(R-CBM), which merge CBMs and GNNs together. To be more specially, it encode atom into concept like CBM and did message passing afterwards like GNN.

**Strengths:**

1. The idea of combining GNN and CBM is novel and it enable the CBM to learn relational data.
2. The results of the experiment are somewhat competitive.
3. Relative clarity in the articulation of the motivation for the experiment

**Weaknesses:**

1. It seems like the CBM provide a initialization of GNN model, therefore, it is not enough to compare with only CBM model as baseline. Because essentially, this is a GNN model. I think we should provide baselines for different GNN + different initialization as comparison.

2. Concepts in CBM are clearly defined. Can the author clearly defined what are the concepts in each of your dataset? When the dataset become larger, how to define your concepts?

3. I think some good property that R-CBM learned like example 3.4 and 3.5 could be formalized better.

4. Table 4 list some easy relations that R-CBM learned. While I think the power of R-CBM should be learning relations that people could not identify. Could you show some other complex relations?

**Questions:**

See the weakness above

---

> ### Author Rebuttal · Authors · 2024-08-06
>
> *Table 4 lists easy relations that R-CBM learned. Could you show complex relations?*
> **Please notice that Table 4 shows rules learnt by R-DCR, other R-CBMs do not learn rules, but rather enable concept interventions (which is the main purpose of concept-based models).** R-DCR is the relational adaptation of Deep Concept Reasoner (DCR) \[1\], which was a method designed to provide **simple** **instance-based rules to aid interpretability** (cf. with section “Rule parsimony” in the original DCR paper \[1\]). In DCR, the complexity of the rules could be controlled with a hyperparameter $\\tau$. Modifying this hyperparameter leads to more complex rules, even though this is in direct opposition with the aim of interpretability. For instance, in RPS we obtained two different equivalent rules tuning this hyperparameter:
> $\\tau=1$: wins(player1) $\\leftarrow$ $\\neg$rock(player1) $\\land$ paper(player1) $\\land$ $\\neg$scissors(player1) $\\land$ rock(player2) $\\land$ $\\neg$paper(player2) $\\land$ $\\neg$scissors(player2)
> which is equivalent but longer than the following:
> $\\tau=100$: wins(player1) $\\leftarrow$ paper(player1) $\\land$ rock(player2)
> In KGs, by tuning this hyperparameter we also found more complex rules with redundant terms such as (notice that neighborOf(X,Y) and locatedIn(Y,W) provide further relevant evidence but are redundant):
> locatedIn(X,Z) $\\leftarrow$ neighborOf(X,Y) $\\land$ locatedIn(Y,W) $\\land$ locatedIn(X,W) $\\land$ locatedIn(W,Z)
> And we also found longer chains such as:
> locatedIn(X,W) $\\leftarrow$ neighborOf(X,Y) $\\land$ neighborOf(Y,Z) $\\land$ neighborOf(Y,Z) $\\land$ locatedIn(Z,W)
>
> *It seems like the CBM provides an initialization of the GNN model & GNN baselines:*
> *   **R-CBMs are not providing the initialization of the GNN model. On the contrary, GNNs provide the initialization of the R-CBM bottleneck.** To clarify the relation between R-CBMs and GNNs consider the following analogy with the non-relational setting. A CBM \[12\] can be seen as composed of three functions $g’ \\circ g’’ \\circ f$: the input encoder $g’: x \\rightarrow h$ (mapping raw features to embeddings), the concept predictor $g’’: h \\rightarrow c$ (mapping embeddings to concepts), and the task predictor $f: c \\rightarrow y$ (mapping concepts to tasks). In the non-relational setting and in the image domain, ResNets are a common function $g’$ in the literature. In the relational setting, our input encoder $g’$ is composed of GNN layers. As a result, R-CBMs do not provide initialization for the GNN, but rather the GNN provides the input for the atom predictor of the R-CBM.
>
> *   **For a fair comparison, we compared R-CBMs with equivalent black-box baselines following the CBM literature.** For instance, in the non-relational settings, the original CBM paper (\[12\]) compared a ResNet (input encoder) + MLP (task predictor) black box baseline with a CBM composed of a ResNet (input encoder) + linear layer (concept predictor) + MLP (task predictor). Following this, in our experiments we compared a GNN (encoder) + MLP (task predictor readout) black-box baseline with a R-CBM composed of GNN (encoder, the same of the black box) + linear atom encoder (concept predictor) + MLP (task predictor readout).
>
> *   **We also provided additional black-box baselines in KGs’ experiments (Table 2 in submitted paper) and we also compared with an additional graph-based method that is not black-box (cf. with C&S in Table A of the attached pdf).**

---

> ### Author Response · Authors · 2024-08-12
>
> Please let us know if you have any further questions or things we could clarify further. If not, we would appreciate if you could consider updating your review based on our replies.

---

### Official Review · Reviewer_CYvd · 2024-07-13

**Soundness:** 3
**Presentation:** 4
**Contribution:** 3
**Rating:** 6
**Confidence:** 3

**Summary:**

The paper introduces Relational Concept Bottleneck Models (R-CBMs), which address the challenge of designing interpretable deep learning models that operate in relational domains. Existing Concept Bottleneck Models (CBMs) are interpretable but lack the capability to manage relational data, while Graph Neural Networks (GNNs) can handle relational data but are not as interpretable. R-CBMs integrate the strengths of both by allowing interpretability in a relational context. They achieve this by mapping input features to a set of human-understandable concepts, then using these concepts to make predictions. The paper evaluates R-CBMs across various experimental settings, demonstrating that they match or exceed the generalization performance of existing relational models, support quantified concept-based explanations, respond effectively to test-time interventions, and perform robustly in challenging scenarios like out-of-distribution testing and limited data availability

**Strengths:**

1. The integration of GNN with CBM is novel.
2. The generalization and efficiency experiment shows impressive result. This is important if we can extend CBM's interpretability to OOD problems.
3. The writing is excellent. Overall this is an enjoyable read. The authors clearly discuss the reasons of each component very clearly. With examples, they clearly articulate the aim of the paper.
4. The experiments were extensive, though real-world imaging datasets were not used.

**Weaknesses:**

1. Related work is insufficient but should have included the interpretable methods to fix this issue. Here are the two papers, the authors should include:
* [Posthoc based]

[1] Dividing and Conquering a BlackBox to a Mixture of Interpretable Models: Route, Interpret, Repeat. Ghosh et al. ICML 2023

[2] POST-HOC CONCEPT BOTTLENECK MODELS. Yuksekgonul et al. ICLR 2023

* [CLIP based]

[1] Label-Free Concept Bottleneck Models. Oikarinen et al.

[2] Visual Classification via Description from Large Language Models. Menon et al. ICLR 2023

* [Relational CBM]

[1] Relational Concept Based Models. Barbiero et al. arXiv 2023.

[2] Interpretable Neural-Symbolic Concept Reasoning. Barbiero et al. ICML 2023.

2. Insufficient baselines. The authors should comapre with atleast one of the posthoc based baselines. Also original CBM is not SOTA anymore as there are multiple variants of CBM.

3. Do the authors assume to have concept annotation? this is expensive. Can it be aliviated with the LLMs to constuct the concepts. There are existing works which include the LLM to deduce the concepts.

4. Currently the community is concerned about the incompleteness of the the concept annotations. So with the assumption of having concept annotation the authors fail to answer this question? How to address this incompleteness?

5. CBMs can be used to find easy/hard samples. With relational model this can be done. The authors can perform an experiment for finding out the easy/hard samples. They can see Route, Interpret, Repeat paper.

6. How to quantitatively evaluate the concepts extracted? The authors should have computed the concept completness scores (Yeh at al.) to do so.

7. How to extend this work for imaging datasets like scene graph understanding?

**Questions:**

See Weaknesses

**Limitations:**

See Weaknesses

---

> ### Author Rebuttal · Authors · 2024-08-06
>
> *Concepts’ evaluations, like e.g. concept completeness:*
> **We report the completeness scores of each concept-based model wrt the relational baseline**, following Equation 1 in \[Yeh, et al.\].  The results are shown in Table D of the attached pdf (we added this result in Table 6, Section 5). An evaluation of concept efficiency was already in Table 5 in the submitted paper (showing the impact of reducing the number of concept/task supervisions during training).
>
> *Can R-CBMs be used to find easy/hard samples?*
> **Yes, R-CBMs can be used for this.** The methodology presented in the “Route, Interpret, Repeat” paper is a technique which is not specific to a particular CBM, and it can be adapted to our methodology as well. In our framework, we consider as hard examples the ones whose prediction is highly uncertain when using a CBM with a propositional template (see CBM-Deep rows in Table C of the attached pdf). When using a relational template, instead, we verified that the distribution of the prediction uncertainty significantly decreases. We show this in Figure A (attached pdf) where the prediction uncertainty decreases when transitioning from a propositional to a relational template. Table C (attached pdf) shows the concept/task activations for the hardest example to classify using the propositional template (high uncertainty) and the corresponding predictions when using the relational template (low uncertainty). We added this analysis Appendix A.6 of the revised version of the paper.
>
> *References for related works:*
> **We thank the reviewer for the suggestions, we added the missing relevant references** (with the only exception for “Interpretable Neural-Symbolic Concept Reasoning” \[1\] which was already part of our evaluation). Regarding post-hoc and CLIP-based CBMs as related works, we are well aware of these papers, but in our opinion these are not closely related works as they focus on ways of obtaining concept labels when such concept annotations are not available, and their extension to the relational case is not trivial, but requires significant further research as discussed in the common answer to the reviewers.
>
> *How to extend this work for imaging datasets like scene graph understanding?*
> **The “Tower of Hanoi” dataset we used in our experiments already represents an example of a setting similar to scene graphs**, where disk images correspond to objects and their relations (i.e., top(u, v), larger(u, v)) correspond to concepts.
>
> **References**
> \[Yeh, et al.\] Yeh, Chih-Kuan, et al. "On completeness-aware concept-based explanations in deep neural networks." Advances in neural information processing systems 33 (2020): 20554-20565.

---

> > ### Comment · Reviewer_CYvd · 2024-08-12
> > **Post rebuttal comment**
> >
> > I would like to thank the reviewer for the rebuttal. My questions are mostly answered except the LLM and incompleteness.
> > Recent publications aim solve this:
> > [1] A Textbook Remedy for Domain Shifts: Knowledge Priors for Medical Image Analysis. Yang et al.
> > [2] CONCEPT BOTTLENECK MODELS WITHOUT PREDEFINED CONCEPTS. Schrodi et al.
> > [3] LLM-based Hierarchical Concept Decomposition for Interpretable Fine-Grained Image Classification. Qu et al.
> >
> > I would like to remain with my score.

---

> > > ### Author Response · Authors · 2024-08-12
> > >
> > > **We provided the completeness scores of our method and CBM baselines in Table D** (see pdf attached to the global answer to all reviewers) and **explained the reason why concept annotations are often complete by construction in relational domains** (see common answer "Concepts’ definition/annotation for each dataset"). Please let us know if you have further questions regarding incompleteness that we have not addressed.
> > >
> > > **Yes, LLMs might be used to construct concepts, but our paper focuses on an orthogonal research question which is: "how can relational concepts be used in a CBM setting?"**. Moreover, it is still unclear how to perform effective concept-interventions when LLMs are used to generate concepts. Indeed, even in recent papers (including the mentioned "Concept Bottleneck Models Without Predefined Concepts" [published on arxiv the last month]), interventions are limited to intervening on the task predictor's weights and on a cherry-picked selection of samples. Further research might be required to understand whether concept bottlenecks and annotations constructed with LLMs provide the same guarantees in terms of interpretability and intervention effectiveness wrt other CBMs. We consider the integration of LLMs and relational CBMs a wide topic of interest for future works.

---

> ### Comment · Reviewer_CYvd · 2024-08-12
> **Further comments [Reviewer]**
>
> By concept incompleteness, i did not indicate concept completeness score. Concept completeness score indicates how good your concept can be a good predictor of the downstream labels. Concept incompleteness means what if your concept set is incomplete on the first hand. Look at the papers i refered.
>
> Regarding concept completeness score, you provided in the pdf - is those numbers in percentages? usually concept completeness scores are b/w 0-1. I see some numbers are greater than 100 (e.g., 102.52). How is that possible? So, this part i consider is not rightly estimated
>
> For the LLM point, LLM can be used to solve the question of incompleteness. and i believe at this point relational concepts can be obtained by using LLM as well, so these are not orthogonal. How to perform intervention with LLM concepts - there are papers like Language in a bottle (LaBo) (CVPR 2023). However i agree this can be explored in future. But that makes the contribution borderline.
>
> Thanks again.

---

> > ### Author Response · Authors · 2024-08-13
> >
> > Dear Reviewer CYvd,
> >
> > We are sorry for misinterpreting your comment regarding concept incompleteness.
> >
> > **According to Definition 3.1 in Yeh, et al., the completeness score can be higher than 1 (100% in our table), whenever "the best accuracy by predicting the label just given the concept scores" (the CBM's accuracy) is higher than the accuracy of the black box prediction model.** Both the numerator and the denominator are normalized wrt the "accuracy of random prediction to equate the lower bound of completeness score to 0". However, this metric is not normalized to give an upper bound in 1 in its original formulation.
> >
> > Regarding concept incompleteness in the relational domain, consider that in the relational setting the issue of concept incompleteness is less demanding than in the non-relational domain. Indeed, the set of concepts is often complete by construction as both concepts and tasks can relate to the same ground predicates (this is a property of relational datasets, as we described in the example in the common answer "Concepts’ definition/annotation for each dataset"). **One of the main results of our experiments shows that in the nine relational datasets we considered, this is actually the case: relational concept bottlenecks are complete, while non-relational bottlenecks are not.**
> >
> > **Regarding LLMs, we only wanted to point out that this paper aims to fill a different research gap with respect to the problem of incomplete concept annotation and the integration of LLMs with CBMs: to the best of our knowledge, this is the first paper that shows how to construct CBMs in the relational domain**. Our contribution fills this gap notwithstanding whether the set of concepts and their annotations are given from supervisions, extracted from an unsupervised method [2], or provided by an LLM [1,3]. However, we acknowledge that studying concept incompleteness and integrating LLMs with CBMs in relational domains are interesting directions for future research.
> >
> > Thanks again for your feedback and your comments, they definitely helped us improving our work.

---

### Official Review · Reviewer_WYX8 · 2024-07-16

**Soundness:** 3
**Presentation:** 2
**Contribution:** 3
**Rating:** 6
**Confidence:** 4

**Summary:**

This paper introduces Relational Concept Bottleneck Models (R-CBMs), a family of relational deep learning models that can provide some degree of interpretability and explainability; R-CBMs generalise both CBMs and GNNs.

According to the authors, R-CBMs 1) match the generalisation performance of existing black-box models, 2) support the generation of logic explanations, 3) respond to test-time concept and rule interventions, and 4) generalise to out-of-distribution samples and in limited training data regimes.

R-CBMs represent a relational graph as a set of atoms and dependencies among atoms and a directed hypergraph: each hyperedge defines a relational concept bottleneck from several ground atoms to a destination ground atom. Fig. 2 provides an example: the atom p4(b) can be predicted from the atoms [p3(b), p2(a, b), p1(b, a)], and these atoms all belong to the same hyperedge. What is the difference between the hyperedge notation and writing this as a Horn clause, e.g. p4(b) :- p3(b), p2(a, b), p1(b, a)?

R-CBMs are composed of a few components: 1) an atom encoder/predictor, 2) a message-passing component that updates the atom representations based on the dependency hypergraph, and 3) a relational task predictor. Having GNN-like message-passing components for updating the embeddings and the predictions of the atoms, in my opinion, may invalidate the interpretability claims of R-CBMs since GNNs are intrinsically black-box neural models.

One of my concerns in this work is that there are already models that can learn and leverage Horn rules-like structures for interpretable relational predictions, such as Neural Theorem Provers (e.g., https://arxiv.org/abs/1705.11040, https://arxiv.org/abs/2007.06477) -- what's the delta between R-CBMs and NTPs?

NTPs are really similar to the proposed approach -- for example, R-CBMs use "max" as the aggregation function to decide which hyper-edge to use when making a prediction, which is exactly the same strategy used by NTPs for deciding which "proof path" to use to prove whether a given atom is true or not.

Experiments -- probably the most used dataset for relational link prediction in Knowledge Graphs is FB15k-237; why did you use WN18RR instead? Are there scalability issues due to the higher number of atoms or relational predicates in FB15k-237? Also, the paper introducing WN18RR was not cited (as well as several papers introducing the baselines).

For WN18RR, the paper proposes an apparently wide set of baselines. However, extremely simple but effective baselines like ComplEx-N3 (https://github.com/facebookresearch/kbc/, https://arxiv.org/abs/1806.07297, ICML 2018) produce more accurate results than the proposed method (0.58 Hits@10 vs 0.56).

I really loved Tab. 4 with the examples of learn (symbolic) rules; however, learning symbolic rules is something that NTPs can also do -- can you please expand on the delta between R-CBMs and NTPs? Also, it's not completely clear to me how rules are learned -- is it because the encoder/predictor can be used to learn a dependency hypergraph that can be used to learn hyper-edges (the Horn clause-like structures)?

Update after reading the rebuttal: the authors addressed most of my concern, I'm increasing my score!

**Strengths:**

1) Interesting work with some degrees of interpretability/explainability!
2) Results look robust for a non-factorisation-based method (I'm referring mainly to WN18RR since other link prediction datasets tend to be mostly solved/saturated)
3) Wide array of graph learning tasks

**Weaknesses:**

1) It is not clear if it's reinventing Neural Theorem Provers (e.g., https://arxiv.org/abs/1705.11040, https://arxiv.org/abs/2007.06477), which can also be used to learn FOL rules via back-prop
2) Why WN18RR instead of, e.g., FB15k-237? Missing very simple but effective link prediction baselines, e.g. https://arxiv.org/abs/1806.07297 (ICML'18)
3) Writing is a bit opaque -- why use "hyper-edges" to introduce Horn clauses?
4) Given all the GNN-like components inside R-CBMs, are they really interpretable?

**Questions:**

1) Can you clarify how the learning of the symbolic rules happens? Is it by learning a link predictor in the atom encoder module that can be used to extract hyper-edges/Horn clauses?
2) Any answer addressing the "Weaknesses" would be really helpful

**Limitations:**

1) Are there scalability limitations of the proposed model? E.g., would you be able to evaluate on FB15k-237?
2) What if you need more than one application of the rules? NTPs can handle that by applying them recursively

---

> ### Author Rebuttal · Authors · 2024-08-06
>
> *Relations with Neural Theorem Provers:*
> **More scalable variations of NTPs, such as CTP and Minerva, have been proved to be weaker baselines than other baselines (e.g., RNNLogic) that we considered in the experiments, e.g. the results in Table 3 from the RNNLogic paper \[24\]. This is why we decided to not include an older method like NTP, for which we would also miss a comparison on larger KGs, as NTP/CTP do not scale on them.** Please also note that our relational extension of CBMs is more general than NTP, or any other specific rule learner. In fact, CBMs do not necessarily learn rules (see the original paper \[12\]), and rule learning is not necessarily required for interpretability, which is usually assessed via concept interventions (Table 3), as discussed by the original CBM paper \[12\].
> However, delving into more technical details, we claim that the R-CBM framework is more general and scalable than NTP for multiple reasons:
>
> *   R-CBMs are not limited to Horn clauses, for example R-DCR can learn rules with negated terms in the body.
>
> *   NTP has never been applied in classic CBMs setups where inputs are not symbolic but images.
>
> *   NTP has never been used to test interventions, which are an essential element for the interpretability of a CBM system.
>
> *   R-CBM templates can represent all the rules using a (subset of a) specified list of atoms in the body at the same time, and the embeddings will be used to determine which actual rule to instantiate in each given context. This is particularly explicit in R-DCR, where a template can form a FOL rule for a grounding and another rule for another grounding. On the other hand, NTP approach to rule learning is to enumerate all possible rules and let the learning decide which rules are useful. Please note that this approach is not scalable to larger KGs because of the combinatorial explosion of rules when there are many predicates in the dataset. Moreover, the rules in NTP are obtained after training by decoding the parameterized rules, **by searching for the closest representations of known predicates.** This is very different from R-DCR, where the rules use exactly the referred predicates and are transparently executed to get the final predictions in all the training phases (cf. with original DCR paper \[1\]).
> We added a summary of the above discussion in the related works (Section 6) in the revised paper.
>
> *Complex-N3 and FB15k-237:*
> **To strengthen our results, we included Complex-N3 as baseline and added a comparison on FB15k-237 as suggested (see Table B in attached PDF)**. Please note that even if ComplEx-N3 provides competitive results, it is still a black box, and it does not directly support concept-based interventions (the main purpose of concept-based interpretability methods such as R-CBM). We added these results in Table 2 of the revised paper.
>
> *Scalability:*
> **Model inference scales as message passing (O(N\*C) where N is the graph size and C is the size of the largest clique)**. As any other method grounding on a relational domain (such as hyper-GNNs), the graph size N grows as the cartesian product of the domain in the template. As discussed in the limitations and in A.2.1 (please note that a deeper discussion is beyond the scope of this paper), there are effective heuristics to limit the size of the graph, while retaining the relevant information to solve a task, see for example \[24,33\].
>
> *Can you clarify how the learning of the symbolic rules happens?*
> **R-CBMs do not aim to learn rules, but rather to enable concept interventions (which is the main purpose of concept-based models) as discussed in the original CBM paper \[12\].** However, as R-CBMs generate symbolic concept layers, existing rule-based approaches can be applied on this layer to make interpretable predictions. In our paper, we used and compared with existing rule-based approaches including concept-based (DCR, \[1\]), NeSy-based (DeepStochLog), and KG-based (RLogic, RNNLogic, etc). Rule learning depends on the chosen method as described in the original papers respectively. For instance, DCR (and R-DCR, its relational adaptation) consists of (i) a neural module to learn the rule structure (by learning the relevance and polarity of each concept) and (ii) a symbolic module to execute the rule on the predicted concept truth values to produce the final prediction.
>
> *What if you need more than one application of the rules?*
> In R-CBMs the recursive application of the rules corresponds to repeating message-passing operations (Sec 3.2, L121).
>
> *Why use "hyper-edges" to introduce Horn clauses?*
> The hyperedge notation is more natural in the GNN community, where the current literature describes message passing operations on hypergraphs \[Feng et al.\].
>
> *References for dataset/baselines:*
> We have added to the revised paper the references when the dataset/baseline (such as WN18RR) is first mentioned in the paper.
>
>
> **References:**
> \[Feng et al.\] Feng, Yifan, et al. "Hypergraph neural networks." Proceedings of the AAAI conference on artificial intelligence. Vol. 33. No. 01. 2019.

---

> > ### Comment · Reviewer_WYX8 · 2024-08-09
> > **Thanks!**
> >
> > Thanks, I will increase my score!

---

### Author Rebuttal · Authors · 2024-08-06

**Answer to all reviewers and ACs**
--------------------------------------

We first thank the reviewers for their thoughtful and insightful feedback. We think that by working on their comments, the quality of our manuscript has certainly improved, and we hope to have addressed all the raised concerns in this rebuttal. We reply to questions shared by two or more reviewers in this comment and reply to specific questions of single reviewers in the comments under their respective feedbacks.

Summary of Changes
---------------------

In the revised version of the paper, we have included both the results of some additional experiments we conducted during the rebuttal and a few lines to better clarify some insightful points raised by reviewers. **However, the core of our work’s contribution and evaluation remains unchanged**. In the following we summarise the list of changes we have made. **Throughout the whole rebuttal we use references with letters (A-D) for tables and figures in the additional pdf page in attachment, while references with numbering refers to our original paper.** The changes are the following:

1.  We added a new experiment to compare with the Correct and Smooth (C&S) method (Table A, @XAga).

2.  We added a new experiment on the dataset FB15k-237, and added Complex-N3 among the baselines for the link prediction task (Table B, @WYX8).

3.  We added an experiment to show how our framework can be also used to identify easy/hard samples in a dataset (Table C, @CYvd).

4.  We included the completeness score of the concepts used by R-CBM (Table D, @raJm,@CYvd).

5.  We included a discussion on NTP (and its extensions) in the related work section (@WYX8).

6.  We added an example in Section 4 to clarify how concepts and tasks are defined in all datasets (@raJm, @CYvd).


**\# Answer to common questions**
---------------------------------

*@XAga @CYvd @WYX8–Baselines:*
**Our work already includes SotA baselines** including SotA concept-based models (Deep Concept Reasoner \[1\], which is a further evolution of Concept Embedding Models \[5\], which is a SotA CBM architecture), SotA relational reasoners (DeepStochLog using optimal, ground-truth rules), and SotA KGEs (e.g., RNNLogic and the newly added ComplEx-N3).

*   Regarding relational post-hoc CBMs, they would be interesting to consider, but as far as we know, there are no papers showing how to extend post-hoc CBMs to the relational case (as far as we know we are the first authors to do this extension for standard CBMs in general). In this setting, post-hoc CBMs would require relational concept discovery. However, CAV (method used in post-hoc CBMs to find concept vectors) is not designed for relational settings. In our view, this would require non-negligible further research that is not directly related to the objective of our paper.

*   We have added (in Table 1 of the revised paper) the results of the suggested GNN baseline (Correct and Smooth) on our splits of the graph benchmarks (Cora, Citeseer, Pubmed) using the same GNN backbone used for all other methods (see Table A of attached PDF).


*@XAga, @WYX8–Impact of message passing, task predictor, and aggregation on interpretability:*
**The meaning of 'interpretability' that we adhere to in this paper aligns with the standard notion used in CBMs: “Interventions make concept bottleneck models interpretable in terms of high-level concepts” (Koh et al. \[12\]) (quantitatively evaluated in Table 3)**. Hence, we notice that interpretability in CBMs (as well as R-CBMs) is not necessarily related to the transparency of the task predictor, but rather depends on the fact that both the input and the output of the task predictor are interpretable units of information (e.g., concepts and tasks). Similarly to ResNets in the non-relational case, GNNs are considered black boxes because message-passing layers are usually mappings between non-interpretable features (e.g. raw features or embeddings). However, R-CBMs’ message-passing propagates interpretable concepts. This enables to enlighten all message passing steps (over possibly multiple iterations), similarly to how concepts enable to enlighten the decision process of the task predictor in non-relational CBMs. In our experiments, we also considered fully-transparent task predictors based on DCR, which learns logic rules to produce task predictions based on concept activations (cf. L198-208 and paragraph “Models” in Section 4 and A.4).

*@raJm, @CYvd–Concepts’ definition/annotation for each dataset:*
**The concepts’ definitions for RPS and Hanoi are reported in Appendix A.1. In all the other datasets there is no explicit distinction between the set of concepts and task predicates/atoms**. In order to further clarify this, we added in Section 4 the following example: “Let us consider as an example the Countries dataset. The task “locatedIn(France, Europe)” could be inferred by the concepts “locatedIn(Italy, Europe)” and “neighborOf(Italy, France)”, i.e. “locatedIn(France, Europe) $\\leftarrow$ locatedIn(Italy, Europe) $\\land$ neighborOf(Italy, France)”. But at the same time, “locatedIn(France, Europe)” could work as a concept to predict the task “locatedIn(Spain, Europe)” in another inference step, i.e. “locatedIn(Spain, Europe) $\\leftarrow$ locatedIn(France, Europe) $\\land$ neighborOf(France, Spain)”. This shows that the same predicate (locatedIn), and possibly the same ground atom (locatedIn(France, Europe)), can be used both for concepts and for tasks. As a result, there is not any additional cost to annotate concepts as these are the same labels already present in the original dataset.”.

---

### Decision · Program_Chairs · 2024-09-25

**Decision:**

Accept (poster)

**Comment:**

The paper introduces Relational Concept Bottleneck Models (R-CBMs). Roughly speaking, R-CBMs merges CBMs and GNNs into an interpretable relational setting. The reviewers agree that this is novel and interesting work, while also pointing our some downsides, see e.g. the discussion about "concept incompleteness" and adding simple GNN-based models as additional baselines. In any case, the rebuttal of the authors helped to clarify some of the issues, and the reviews lean towards accept. I fully agree.  It would be nice if you could clarify the difference to "relational bottlenecks" (https://arxiv.org/pdf/2309.06629).